# Study on Tribological Characteristics of Textured Surface under Convergent Oil Film Gap

**Guangyao Bei** , **Chenbo Ma \***, **Xilong Wang, Jianjun Sun and Xingya Ni**

College of Mechanical and Electronic Engineering, Nanjing Forestry University, Nanjing 210037, China
\* Correspondence: machenbo@njfu.edu.cn

**Abstract:** Under the condition of convergence, a textured non-parallel 2D slider bearing model was established, and the tribological properties of textured surface under the convergence gap were numerically studied with the load-carrying capacity as an indicator. Firstly, whether the convergence ratio parameter can accurately characterize the joint effects of film thickness difference and oil film thickness on the surface tribological properties was verified, and the effects of film thickness difference and oil film thickness on the load-carrying capacity of textured and non-textured surface were studied, respectively. The results show that the efficiency of improving the load-carrying capacity of the surface structure first increases and then decreases with the increase of the oil film thickness. In the case of large film thickness difference, the surface texture will reduce the efficiency of improving the load-carrying capacity. In addition, the effects of texture depth, texture width, and sliding velocity on the load-carrying capacity under the convergence gap are also studied. In particular, an optimal texture width to maximize the load-carrying capacity exists.

**Keywords:** non-parallel surface; surface texture; convergence ratio; load-carrying capacity

## 1. Introduction

Surface microtexture technology is to process regularly arranged convex or concave structures on the contact surface to improve the tribological properties of various application surfaces [1]. It has been widely used in bearings, biomaterials, metalworking, and so on [2–4]. In 1966, Hamilton et al. [5] first found that machining micro pits on the surface of friction pairs can effectively improve their friction characteristics, and proposed the hydrodynamic effect. Anno et al. [6] further proved by experiments that the micro convex on the face seal and thrust bearing surface can provide better lubrication performance. After that, Etsion et al. [7,8] made pioneering research on the application of surface texture on the end face of mechanical seals, which confirmed that surface texture is an excellent way to improve the tribological performance of mechanical parts. It makes the application of surface texture with micro scale characteristic size on the friction surface widely concerned by the tribology community.

The geometric parameters of surface texture are important factors affecting tribological properties. Many studies have shown that only the surface texture with appropriate parameters can play its optimal role in surface modification [9]. Among them, the texture width and texture depth will have a great impact on the tribological behavior of textured surface [10]. Wang et al. [11] studied the friction force on the annular texture surface under different geometric parameters, and the results show that the width of the annular texture has a significant impact on the wear reduction effect. Rao et al. [12] studied the tribological properties of textured surface friction pairs under the conditions of different groove widths through experiments, and found that the texture with a width of 2 mm is more suitable for low-speed conditions, and the texture with a width of 3 mm is more conducive to improve the wear performance under high-speed conditions. Li et al. [13] conducted multi-objective optimization on texture structure and distribution parameters

to improve the tribological performance of thrust bearings, and found that texture width is the most sensitive parameter to bearing capacity and friction. Zhang et al. [14] studied the tribological properties of textured stainless steel surfaces with different texture depths, and found that the effectiveness of surface texture largely depends on the depth of texture. After that, Niu et al. [15] conducted friction and wear experiments on textured carbon steel surfaces with different geometric parameters, and pointed out that texture depth is the main factor affecting friction characteristics. Li et al. [16] prepared square matrix dimple texture surfaces with different geometric parameters on the surface of GCr15 steel. When the texture depth is 10 mm, the area density is 8%, and the texture diameter is 100 mm; it has the best lubrication performance. Traore et al. [17] analyzed the influence of flow velocity on the friction characteristics of textured surface and found that the friction coefficient decreased significantly with the increase of flow velocity. Wang et al. [18] studied the characteristics of fractal textured surface under mixed lubrication and concluded that the friction coefficient of textured surface will decrease with the increase of sliding speed.

However, a completely parallel surface does not exist. Under the application condition, the two surfaces are often in a non-parallel state due to manufacturing and installation errors, surface stress, and heating or installation needs. Compared with parallel surface contact, there are few studies on the non-parallel application of surface texture [19]. Malik et al. [20] studied the load-carrying capacity of a parallel slider bearing and an inclined slider bearing with a textured surface. They concluded that the inclined slider always has a better load-carrying capacity than the parallel slider. Muthy et al. [21] used the optimal texture parameters under the condition of parallel sliding blocks to study the plane inclined slider, and found that there was a critical value of the tilt angle affecting the average pressure. Dobrica et al. [22] conducted a numerical analysis on the influence of texture in the inclined slider, and found that, in the convergent plane inclined slider, the full texture will have a negative impact on the friction performance, while part of the texture can improve the performance.

Furthermore, the researchers introduced the convergence ratio parameter to characterize the degree of convergence of the textured surface in the study of non-parallel textured surface, so as to study the tribological characteristics of the textured surface under the convergent oil film gap more easily. Cupillard et al. [23] analyzed the flow of inclined slider bearings under different convergence ratios and different texture depths. The results show that, when the convergence ratio is small, the load-carrying capacity efficiency of texture surface is higher. Papadopoulos et al. [24] studied the optimal texture parameters of a textured three-dimensional inclined slider by using a genetic algorithm. They found that the efficiency of texture would decrease with the increase of convergence ratio. In subsequent research [25], they found that the increase of convergence ratio would increase the stiffness and reduce the damping of the bearing. Zhang et al. [26] optimized the coverage of the circular texture on the bearing slider to improve its tribological performance, and took the convergence rate and width as indicators to ensure that the slider with a smaller convergence rate and width has higher tribological characteristics. Rosenkranz et al. [27] studied the friction performance of grain convergent bearing under full film lubrication under the condition of thick oil film through experiments. Texture helps reduce friction at high convergence rates. Meanwhile, in the case of low convergence rate, the texture at the bearing inlet will significantly reduce friction. The influence of convergence ratio on the tribological properties of textured surface can be attributed to the comprehensive influence of oil film thickness and tilt angle on the tribological properties of the textured surface. Although many scholars have studied the influence of convergence ratio on the tribological properties of textured surface according to different application backgrounds and reached some conclusions, whether this parameter can accurately characterize the joint influence of the two factors on the tribological properties is still lacking the corresponding verification.

Thus, the tribological characteristics of the textured inclined 2D sliding bearing surface under the condition of convergence are numerically studied by these main findings, and the rationality of the convergence ratio parameter is analyzed and verified. In addition, the

difference of load-carrying capacity between textured surface and non-textured surface is also compared to verify the effectiveness of texture under the condition of the convergence gap. In addition, the effects of texture depth, texture width, and sliding velocity on the tribological properties of textured non-parallel surfaces were systematically studied.

## 2. Establishing the Model

The non-parallel oil film gap is used as the research object to study the tribological properties of textured surfaces under the convergent gap. The geometric model of textured non-parallel surface is shown in Figure 1. The total length $L$ of the computational fluid domain is 1000 μm. The width of the pit in the texture area is $d$, the depth of the pit is $h_p$, the oil film thickness at the small end of the model is $h_0$, and the oil film thickness at the large end is $h_1$. The value of tilt angle $\theta$ is determined by the oil film thickness difference between the inlet and the outlet, expressed as follows:

$$\tan \theta = (h_1 - h_0)/L \tag{1}$$

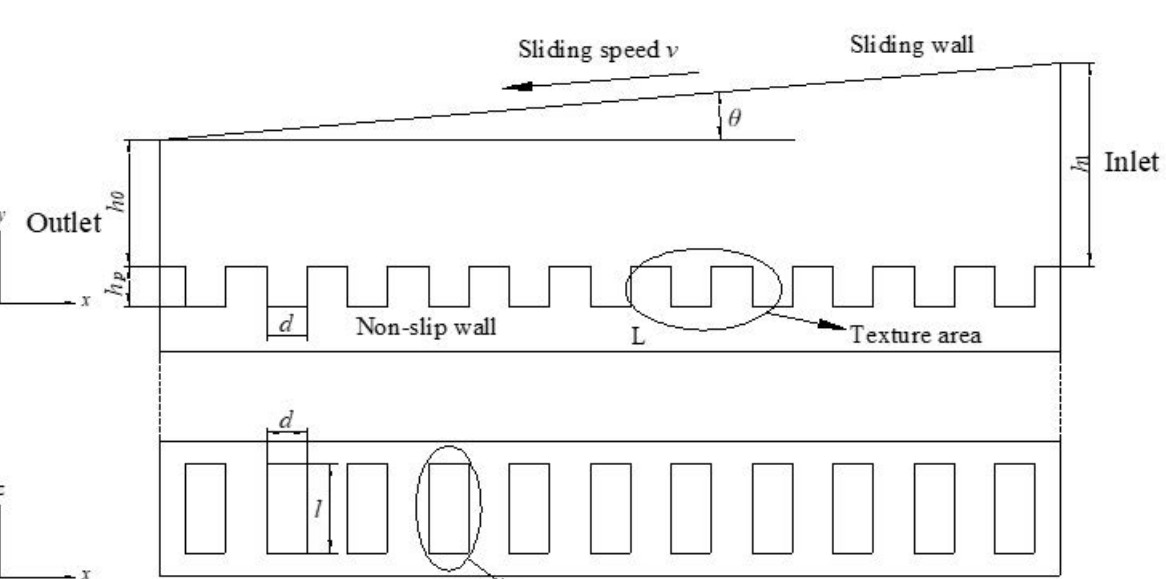

**Figure 1.** Textured non-parallel surface lubrication model.

The inlet and outlet oil film thickness difference $\Delta h$ is used to characterize the influence of the tilt angle for the convenience of follow-up research:

$$\Delta h = h_1 - h_0 \tag{2}$$

### 2.1. Theoretical Basis

The Navier–Stokes (N–S) equation is a motion equation used to describe the momentum conservation of viscous incompressible fluid [28]. This study is based on the N–S equation. The 2D textured non-parallel surface symmetrically distributed along the z plane is taken as the research object and solved by FLUENT software (18.0, ANSYS, Canonsburg, PA, USA).

The continuity equation based on mass conservation is shown in Equation (3) [29]:

$$\nabla \cdot (\rho V) = 0 \tag{3}$$

Equations (4) and (5) show the expressions of the obtained 2D momentum equation along the X and Y directions, respectively:

$$\nabla \cdot (\rho u V) = -\frac{\partial p}{\partial x} + \frac{\partial \tau_{xx}}{\partial x} + \frac{\partial \tau_{yx}}{\partial y}, \tag{4}$$

$$\nabla \cdot (\rho v V) = -\frac{\partial p}{\partial y} + \frac{\partial \tau_{xy}}{\partial x} + \frac{\partial \tau_{yy}}{\partial y}. \tag{5}$$

In Equations (2) and (3):

$$\tau_{xx} = \lambda(\nabla \cdot V) + 2\mu\frac{\partial u}{\partial x}, \tag{6}$$

$$\tau_{yy} = \lambda(\nabla \cdot V) + 2\mu\frac{\partial v}{\partial y}, \tag{7}$$

$$\tau_{xy} = \tau_{yx} = \mu\left(\frac{\partial v}{\partial x} + \frac{\partial u}{\partial y}\right), \tag{8}$$

$$\lambda = -\frac{2}{3}\mu. \tag{9}$$

where $\rho$ indicates the density of the lubricating medium; $u$ and $v$ are the velocity component in the $x$ and $y$ directions, respectively; $p$ is the lubricating film pressure; $t$ is the time; $\tau$ is shear stress; $\mu$ is the viscosity coefficient; and $\lambda$ is the second viscosity coefficient.

*2.2. Meshing*

The mesh independence is verified before determining the mesh generation model to ensure the optimization of calculation speed and solution accuracy. Given that the model in this study differs by two orders of magnitude in the length direction of the fluid domain and the size of the oil film thickness, the grid independence is impossible to verify using the number of grids. Accordingly, the independence of the grid size in the direction of the flow field and the grid size in the direction of the oil film thickness must be verified. The grid independence is verified by the smooth convergent oil film gap when $h_0 = 10$ μm and $h_1 = 11$ μm.

(1) Verification of grid independence of the oil film thickness direction

The grid size of the fixed flow field direction is 0.5 μm. The grid dimensions in the oil film thickness direction are taken as follows: 2, 1, 0.8, 0.6, 0.4, 0.2, 0.1, and 0.05 μm. Table 1 shows the grid number and load-carrying capacity corresponding to the model with load-carrying capacity as the measurement index to analyze the independence of the grid size.

**Table 1.** Grid size in the film thickness direction, number of corresponding grids, and load-carrying capacity.

| Parameter | Value | | | | | | | |
|---|---|---|---|---|---|---|---|---|
| Grid size in film thickness direction (μm) | 2 | 1 | 0.8 | 0.6 | 0.4 | 0.2 | 0.1 | 0.05 |
| Number of grids | 14,014 | 24,024 | 28,028 | 38,038 | 54,054 | 104,104 | 204,204 | 404,404 |
| Load-carrying capacity of the upper wall (N) | 456.13307 | 479.28122 | 481.81329 | 484.74261 | 486.32495 | 487.35166 | 487.79908 | 487.70675 |

When the grid size in the oil film thickness direction is large, the variation of the load-carrying capacity is obvious. When the number of grids is greater than 104,104, the grid size in the corresponding oil film thickness direction is less than 0.2 μm. The load-carrying capacity fluctuates in a small range, and the influence can be ignored.

(2) Verification of the grid independence of the flow field direction

The grid size in the direction of the fixed oil film thickness is 0.1 μm. The grid dimensions in the flow field direction are taken as follows: 2, 1.5, 1, 0.5, 0.3, and 0.1 μm. Table 2 shows the corresponding number of grids and the load-carrying capacity of the upper wall.

**Table 2.** Flow field direction grid size and number of corresponding grids.

| Parameter | Value | | | | | |
|---|---|---|---|---|---|---|
| Grid size in flow field direction (μm) | 2 | 1.5 | 1 | 0.5 | 0.3 | 0.1 |
| Number of grids | 51,204 | 68,238 | 102,204 | 204,204 | 340,170 | 1,020,204 |
| Load-carrying capacity of the upper wall (N) | 488.33234 | 488.10609 | 487.7079 | 487.60867 | 487.54969 | 487.22866 |

In the process of changing the grid size in the flow field direction from 0.1 μm to 2 μm, the variation of the oil film load-carrying capacity is less than 0.023%. Therefore, the change in the load-carrying capacity is independent of the grid size in the flow field direction.

The grid size of the textured non-parallel surface lubrication model is 0.2 μm along the oil film thickness direction and 0.5 μm along the flow field direction based on the verification of grid independence to make the grid aspect ratio in the oil film thickness direction and flow field direction within an appropriate range, combined with the variation trend of load-carrying capacity under different grid sizes in the flow field direction.

*2.3. Boundary Conditions and Solver Parameter Settings*

In Figure 1, the inlet and outlet of the oil film gap are set as the pressure boundary conditions, and the pressure value is 0 Pa. The lower wall is a fixed surface and set as a non-slip boundary condition. The upper wall is set as the moving surface with sliding speed $v$ = 20 m/s, and the direction is determined to satisfy the convergence condition.

Mechanical lubricating oil is used as the lubricating medium, and the lubricating fluid is assumed to be an incompressible fluid. The selected solver is pressure-based, and the time type is set to steady-state. For the setting of cavitation, a mix multiphase flow model is adopted; phase 1 is set as liquid phase, and phase 2 is set as gas phase; the Schnerr–Sauer cavitation model is selected, and the cavitation pressure is set as 30,000 Pa [30]. PRESTO is the selection format of the pressure discrete term, QUICK is selected for the momentum term, and the default relaxation factor of the system is used. The residual accuracy is set to $10^{-7}$, which refers to the convergence accuracy of numerical simulation, that is, the difference between the next step and the previous step is less than the set residual accuracy, and the simulation of load-carrying capacity can be regarded as convergence. The settings of other parameters are shown in Table 3. The load-carrying capacity is taken as the evaluation parameter. Moreover, the ratio of the load-carrying capacity between the textured and the non-textured surfaces is defined as the dimensionless load-carrying capacity to characterize the tribological performance increment brought by texture and more intuitively describe the effectiveness of texture.

**Table 3.** Solver parameter settings.

| Parameter | Value |
|---|---|
| Fluid domain length $L$ (μm) | 1000 |
| Texture unit length $l$ (μm) | 100 |
| Texture width $d$ (μm) | 50 |
| Texture depth $h_p$ (μm) | 5 |
| Sliding velocity $v$ (m/s) | 20 |
| Lubricating oil density $\rho_1$ (kg/m$^3$) | 960 |
| Dynamic viscosity of lubricating oil $\mu_1$ (Pa·s) | 0.048 |
| Gas phase density $\rho_2$ (kg/m$^3$) | 10.95 |
| Gas phase dynamic viscosity $\mu_2$ (Pa·s) | $7 \times 10^{-6}$ |

### 3. Rationality Verification of the Convergence Ratio Parameters

Some researchers have studied the tribological properties of the textured non-parallel surfaces by introducing the parameter of convergence ratio $K$. The calculation formula is shown as follows [23]:

$$K = \frac{h_1}{h_0} - 1 = \frac{h_1 - h_0}{h_0} = \frac{\Delta h}{h_0} = \frac{L \tan \theta}{h_0} \tag{10}$$

The value of convergence ratio $K$ is determined by the tilt angle $\theta$ and the trailing film thickness $h_0$. It can be seen that, for the convenience of research, the previous studies used the case when texture was not introduced, and defined the convergence ratio parameter from a dimensionless point of view, as shown in Equation (10). According to Equation (10), the convergence ratio $K$ is also determined by $h_1$ and $h_0$. Therefore, the same convergence ratio $K$ can be constructed by changing the oil film thickness $h_0$ and oil film thickness difference $\Delta h$ used to characterize the influence of the tilt angle. However, due to the influence of texture on the distribution of oil film thickness after the introduction of texture, it is worth discussing whether the change law of convergence ratio parameter can represent the change of $h_0$ and $\Delta h$.

The difference of the tribological properties under the same convergence ratio can be analyzed to verify whether the convergence ratio parameter can accurately represent the common influence of the two factors.

The values of convergence ratio $K$ are taken as 0.005, 0.01, 0.02, 0.04, 0.06, 0.08, and 0.1 using the textured convergence model. Take the length of texture unit $l$ as 100 μm, rectangular pit width $d$ is 50 μm, and texture depth $h_p$ is 5 μm. Table 4 shows that the trailing film thickness $h_0$ remains unchanged, and the corresponding convergence ratio is constructed by changing the oil film thickness difference $\Delta h$. Table 5 shows that the tilt degree $\Delta h$ remains unchanged, and the corresponding convergence ratio is constructed by changing the trailing film thickness $h_0$.

**Table 4.** Keep trailing film thickness unchanged and change oil film thickness difference.

| Parameter | Value | | | | | | |
|---|---|---|---|---|---|---|---|
| Convergence ratio $K$ | 0.005 | 0.01 | 0.02 | 0.04 | 0.06 | 0.08 | 0.1 |
| Trailing film thickness $h_0$ (μm) | 10 | 10 | 10 | 10 | 10 | 10 | 10 |
| Oil film thickness difference $\Delta h$ (μm) | 0.05 | 0.1 | 0.2 | 0.4 | 0.6 | 0.8 | 1 |

**Table 5.** Keep oil film thickness difference unchanged and change trailing film thickness.

| Parameter | Value | | | | | | |
|---|---|---|---|---|---|---|---|
| Convergence ratio $K$ | 0.005 | 0.01 | 0.02 | 0.04 | 0.06 | 0.08 | 0.1 |
| Trailing film thickness $h_0$ (μm) | 40 | 20 | 10 | 5 | 3.33 | 2.5 | 2 |
| Oil film thickness difference $\Delta h$ (μm) | 0.2 | 0.2 | 0.2 | 0.2 | 0.2 | 0.2 | 0.2 |

Figure 2 shows the variation of the load-carrying capacity of the textured surface with the convergence ratio when changing the film thickness difference and oil film thickness. In both cases, the load-carrying capacity will increase with the increase in convergence ratio, but the change degree is different. When the convergence ratio is small, the textured surfaces in the two cases show similar tribological characteristics. The load-carrying capacity greatly increases with the increase in convergence ratio by changing the oil film thickness, and the increasing rate increases with the increase in convergence ratio. Meanwhile, the load-carrying capacity slightly increases with the increase in convergence ratio by changing the trailing film thickness.

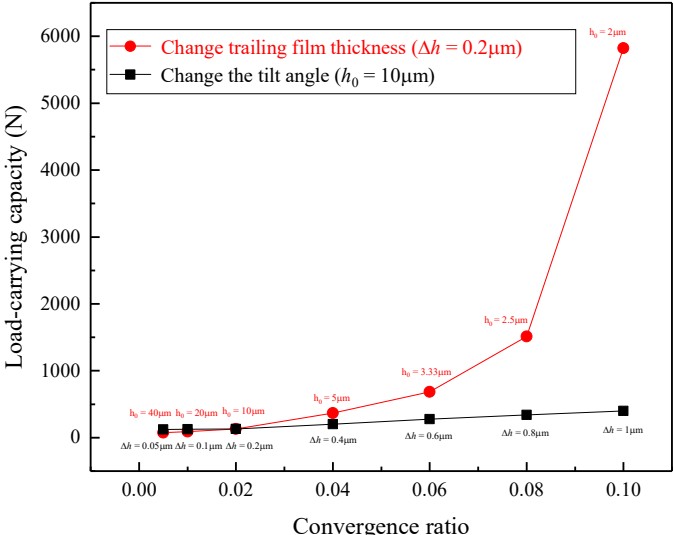

**Figure 2.** Trend of loading capacity with the convergence ratio under texture.

Keeping the same convergence ratio, the load-carrying capacity has the same change trend with the change of oil film thickness difference and oil film thickness. However, the change degree is quite different under the influence of the two factors. Therefore, the convergence ratio parameter is irrational. The differences of convergence ratio and tribological properties under different oil film thickness and oil film thickness difference are studied through the analysis of the different convergence ratio methods obtained to more accurately analyze the rationality of the convergence ratio parameter.

Figure 3 shows the variation law of the load-carrying capacity with convergence ratio under different oil film thickness. Under the condition of the same convergence ratio, the load-carrying capacity corresponding to different oil film thickness has great differences. The load-carrying capacity under different oil film thickness increases with the increase in convergence ratio, but its increasing trend becomes more obvious with the decrease in oil film thickness.

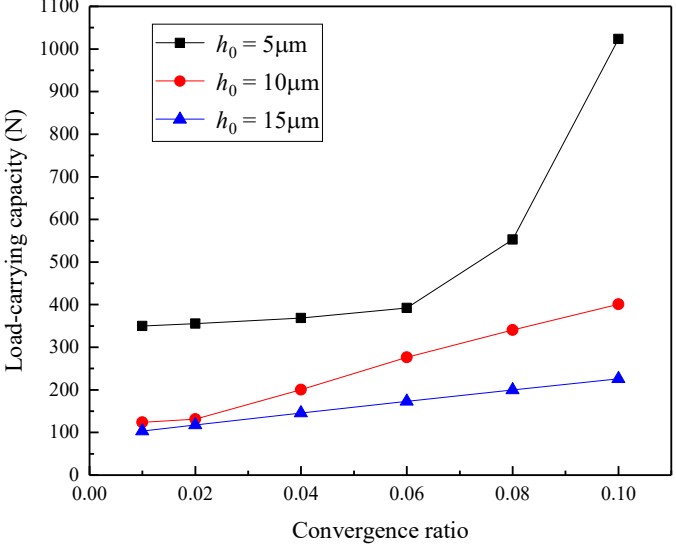

**Figure 3.** Trend of loading capacity with convergence ratio under different trailing film thicknesses.

Figure 4 shows the variation law of the load-carrying capacity with convergence ratio under different film thickness differences. The load-carrying capacity under different film

thickness differences increase with the increase in convergence ratio, and the increasing trend will become more obvious with the decrease in the film thickness difference.

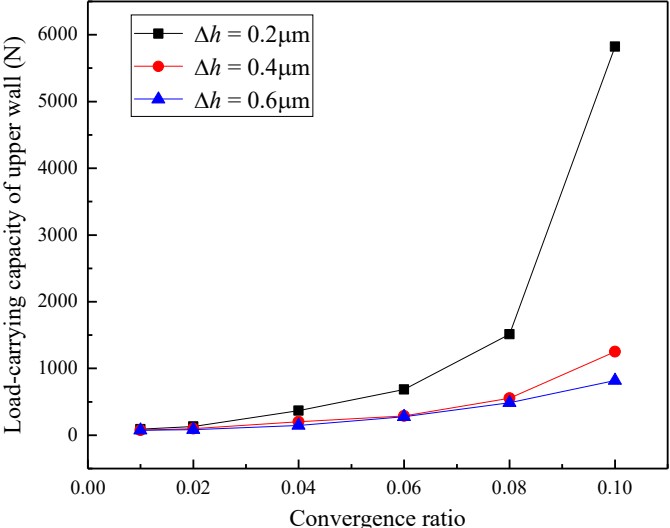

**Figure 4.** Trend of loading capacity with convergence ratio at different film thickness differences.

The result shows that the load-carrying capacity increases with the increase in convergence ratio under the conditions of different oil film thicknesses and film thickness differences. However, the increasing trend of load-carrying capacity will also change when these two parameters change, thereby showing that the influence of convergence ratio affected by oil film thickness and film thickness difference on the load-carrying capacity is quite different. This situation further shows that the convergence ratio parameter is unreasonable when studying the tribological properties of the textured surface. The two parameters of oil film thickness and film thickness difference should be independently studied. In addition, texture depth, texture width, and sliding speed also affect the tribological properties of the non-parallel textured surfaces. Therefore, the effects of these parameters on the textured surface must be studied to better analyze its friction characteristics.

## 4. Influence of Film Thickness Difference, Oil Film Thickness, Sliding Velocity, and Texture Structure Parameters on the Load-Carrying Capacity

### 4.1. Influence of Film Thickness Difference

The variation of the oil film load-carrying capacity with a film thickness difference under the condition of convergence is simulated and analyzed, taking the continuous textured model as the research object. The values of the inlet oil film thickness difference $\Delta h$ are 0, 0.05, 0.1, 0.15, 0.2, 0.4, 0.6, 0.8, and 1.0 μm to characterize different film thickness differences. The trailing film thickness $h_0$ was set as 10 μm. The settings of the other parameters are shown in Table 6.

**Table 6.** Parameter setting.

| Parameters | Value |
| --- | --- |
| Fluid domain length $L$ (μm) | 1000 |
| Texture unit length $l$ (μm) | 100 |
| Texture width $d$ (μm) | 50 |
| Texture depth $h_p$ (μm) | 5 |

Figure 5 shows the variation trend of the load-carrying capacity with a film thickness difference. The load-carrying capacity increases with the increase in the film thickness difference. This situation is mainly because the increase in the film thickness difference

will enhance the convergence effect of texture, resulting in a greater dynamic pressure effect and an increase in the load-carrying capacity. Figure 6 shows the variation law of the dimensionless load-carrying capacity with a film thickness difference. When the film thickness difference is small, the load-carrying capacity of the textured surface is greater than that of the non-textured surface. This finding shows that, under the condition of small film thickness difference, texture will improve the loading performance of the oil film, and the lifting efficiency is the highest when the film thickness difference is 0, that is, the surface is parallel. When the film thickness difference is large, the load-carrying capacity of the textured surface is lower than that of the non-textured surface, which means that the surface texture reduces the load-carrying capacity of the oil film.

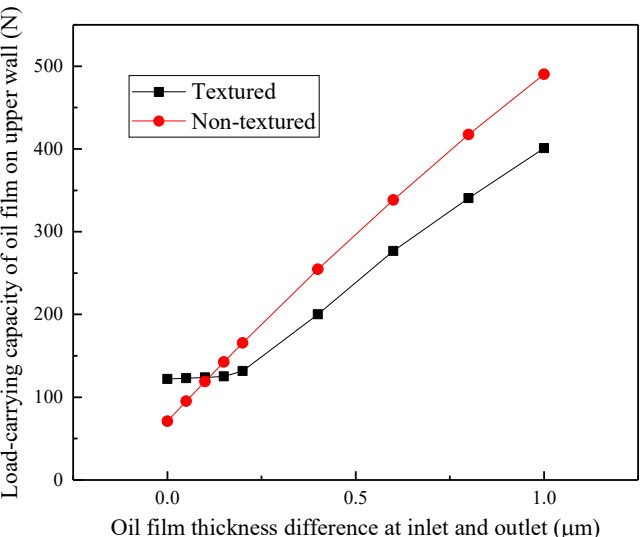

**Figure 5.** Variation trend of loading capacity with the film thickness difference under textured and non-textured conditions.

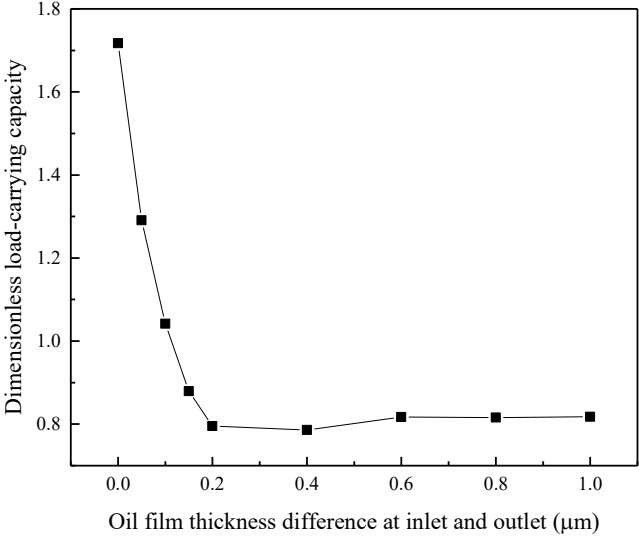

**Figure 6.** Variation trend of dimensionless loading capacity with the film thickness difference.

These phenomena are mainly due to the joint influence of film thickness difference and texture on the pressure of the flow field. Figure 7a,b are the wall pressure distribution curves on the texture surface when $\Delta h$ is taken as 0.05 and 0.6 μm, respectively. The dynamic pressure effect caused by the overall convergence effect at a large film thickness difference is stronger than that at a small film thickness difference regardless of the presence or absence

of texture on the surface, resulting in a greater oil film pressure. Consequently, it has a greater load-carrying capacity at a large film thickness difference. In addition, Figure 7a demonstrates that the maximum pressure of the textured surface is much greater than that of a non-textured surface, but the minimum pressure is not much different. Accordingly, the textured surface obtains an additional load-carrying capacity and verifies the effectiveness of texture in improving the loading performance of the surface under the condition of a small film thickness difference. Figure 7b shows that the difference between the maximum pressure on the textured surface and the maximum pressure on the non-textured surface is only 11%, while that between the minimum and the maximum pressure is 51%. This notion means that the texture will reduce the overall pressure level of the flow field under the condition of a large film thickness difference, showing ineffectiveness.

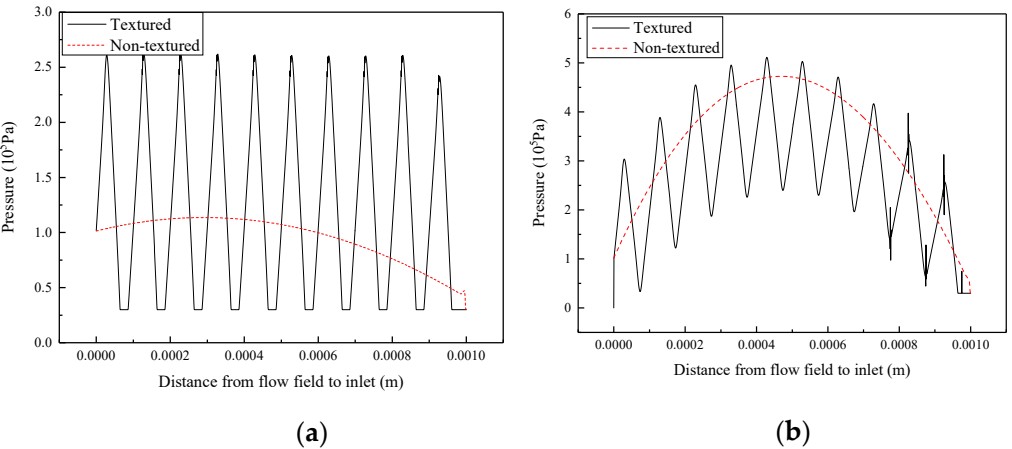

(a)　　　　　　　　　　　　　(b)

**Figure 7.** Pressure distribution curve of upper wall under textured and non-textured conditions. (**a**) $\Delta h = 0.05$ μm, (**b**) $\Delta h = 0.6$ μm.

### 4.2. Influence of Trailing Film Thickness

A textured continuous model is established, and the trailing film thicknesses $h_0$ are taken as 4, 6, 8, 10, 12, 14, 16, 18, and 20 μm. The inlet and outlet oil film thickness difference $\Delta h$ is 0.2 μm. The influence of the oil film thickness on the surface tribological properties under convergence conditions is studied. The other parameter settings are the same as those in Table 6.

Figure 8 shows the variation trend of the load-carrying capacity with oil film thickness under convergence conditions. The load-carrying capacities of the textured surface and non-textured surfaces decrease with the increase in the oil film thickness. This phenomenon occurs because the increase in the oil film thickness will weaken the overall convergence effect when the film thickness difference $\Delta h$ is constant. In addition, when $h_0$ is less than 14 μm, the load-carrying capacity of the non-textured surface is higher than that of the textured surface. This is because, when $h_0$ is small, the oil film has high convergence, resulting in the lack of cavitation area in the contact. With the increase of $h_0$, cavitation may occur in the low convergence contact, resulting in small performance improvement. Figure 9 shows the variation trend of the dimensionless load-carrying capacity with oil film thickness. When the oil film thickness is 13–20 μm, the load-carrying capacity of the textured surface is greater than that of the non-textured surface, increasing by 10% at most. The results show that the oil film thickness will affect the influence of texture on the surface properties.

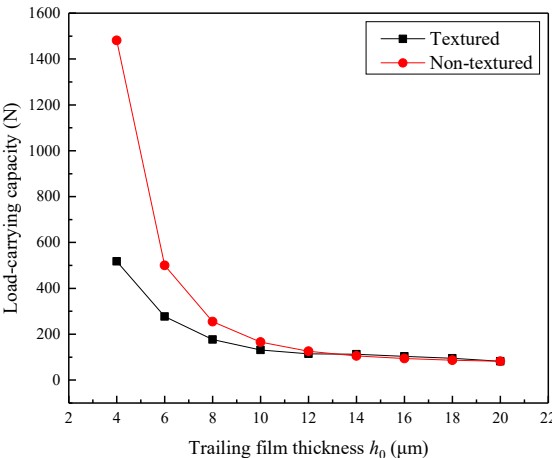

**Figure 8.** Variation trend of loading capacity with trailing film thickness under textured and non-textured conditions.

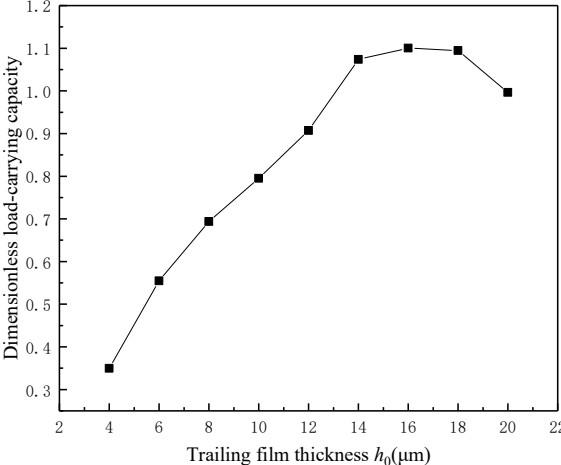

**Figure 9.** Variation trend of dimensionless loading capacity with trailing film thickness.

### 4.3. Influence of Texture Depth

To study the effect of texture depth on the friction properties of the surface under convergent gap, the oil film thickness $h_0$ is taken as 10 μm, and the ratio $h$ of texture depth $h_p$ to trailing film thickness $h_0$ is defined as the dimensionless texture depth. Moreover, the differences $\Delta h$ of the inlet oil film thickness are taken as 0.05, 0.1, 0.2, and 0.4 μm to characterize various film thickness differences. The values of fluid domain length $L$, texture width $d$, and texture unit length l are shown in Table 6. The variation of the load-carrying capacity of the non-parallel texture surface with texture depth under different film thickness differences is studied.

Figure 10 shows the relationship between dimensionless load-carrying capacity and texture depth. When the film thickness difference is large, the value of the dimensionless load-carrying capacity is always less than 1, indicating that texture reduces the loading performance of the oil film. The texture decreases, and the load-carrying capacity of oil film becomes gradually evident with the increase in texture depth. When the film thickness difference is small, the improvement efficiency of the load-carrying capacity first decreases, then increases, and finally decreases with the increase in texture depth. When the texture depth is 7 μm, the load-carrying capacity performance is the best. However, in the case of small film thickness difference, texture does not always improve the load-carrying capacity of oil film, which shows that the depth of texture will affect the load-carrying capacity of the oil film. However, it is not difficult to find from Figure 10 that the dimensionless load-carrying capacity of different texture depths under small film thickness difference

is not all greater than 1, indicating that the effectiveness of texture under the condition of convergent full texture not only depends on the size of film thickness difference, but also is affected by texture depth, such as $\Delta h = 0.05$ μm, the reasonable change of texture depth can increase the load-carrying capacity by about 30%, but, if the texture depth is not properly selected, the minimum dimensionless load-carrying capacity is about 0.93, and the load-carrying capacity is reduced by 7%. It can be seen that it is necessary to reasonably design the texture depth under small film thickness difference.

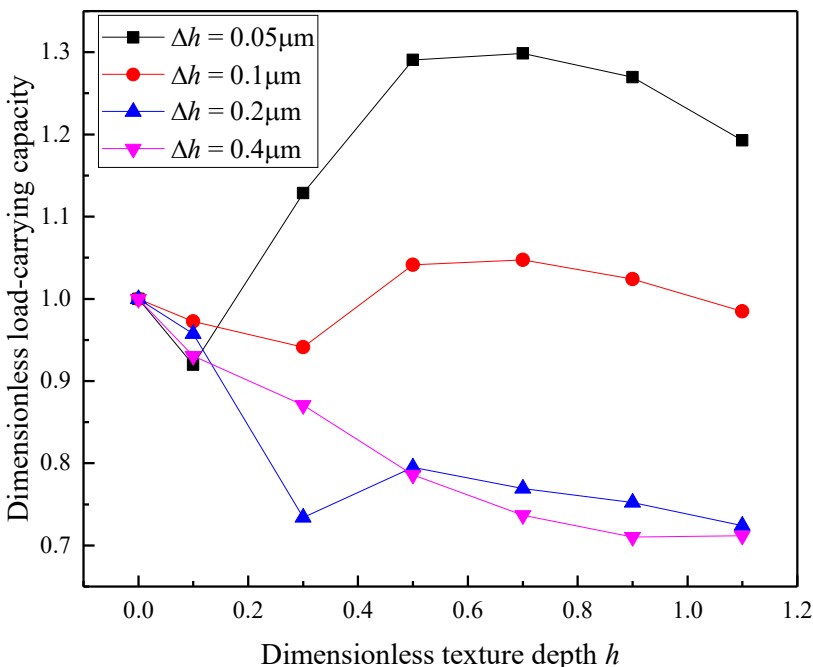

**Figure 10.** Variation of dimensionless load-carrying capacity with dimensionless texture depth.

### 4.4. Influence of Texture Width

To study the effect of texture width on the friction properties of the surface under convergent gap, the dimensionless texture width $w$ is defined as the ratio of texture width to texture unit length. The inlet and outlet oil film thickness difference $\Delta h$ is taken as 0.2 μm. The texture unit length $l$ values are set to 100, 125, 167, 250, and 500 μm. The effect of texture width on the load-carrying capacity of the oil film is studied when the unit length of texture is different under the condition of convergence.

Figure 11 shows the variation trend of the load-carrying capacity with texture width. The load-carrying capacity of the oil film first decreases, then increases, and finally decreases with the increase in texture width, indicating that the presence of an optimal texture width to maximize the load-carrying capacity of the oil film. The reason for this phenomenon may be that, when the texture width is small, it is not easy to produce a dynamic pressure effect. With the increase of texture width, the oil storage capacity increases, the dynamic pressure effect increases, and the load-carrying capacity is improved. As the texture width continues to increase, the effective bearing area of the bearing decreases. Although the texture can produce a dynamic pressure effect, the overall load-carrying capacity decreases. The maximum load-carrying capacity also increases with the increase in texture unit size. Therefore, the loading performance of the oil film is affected by texture width and texture unit size.

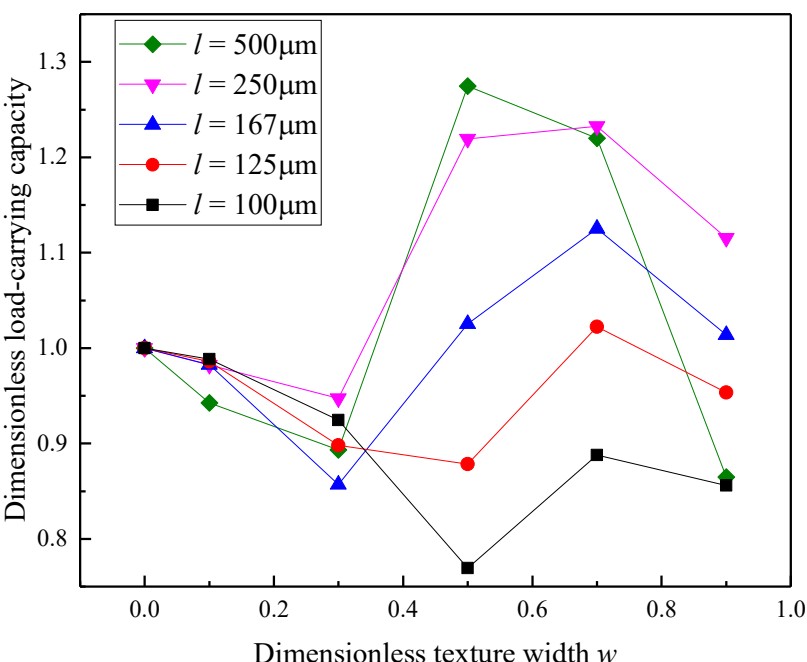

**Figure 11.** Variation of dimensionless load-carrying capacity with dimensionless texture width.

### 4.5. Influence of Sliding Velocity

To study the effect of sliding velocity on the tribological properties, the variation trend of the tribological properties of the textured surface with sliding velocity under different texture depths $h$ (0, 0.1, 0.5, and 0.9) and widths $w$ (0, 0.1, 0.5, and 0.9) was simulated and analyzed.

Figure 12 shows the variation trend of the dimensionless load-carrying capacity with sliding velocity under different texture depths. When the depth of the structure is small, the sliding velocity has little effect on the efficiency of lifting the load-carrying capacity of the textured surface. When the texture depth is large, the improvement efficiency of oil film load-carrying capacity decreases with the increase in the sliding velocity. Figure 13 shows the variation trend of the dimensionless load-carrying capacity with sliding velocity under different texture widths. When the width of the structure is small, the improvement efficiency of the load-carrying capacity performance hardly changes with the change in the sliding velocity. When the texture width is large, the improvement efficiency of the load-carrying capacity first decreases and then increases with the increase in texture width. The reason for this phenomenon is that the effective bearing area of the bearing decreases and the overall load-carrying capacity decreases when the texture width is large. With the increase of sliding velocity, the oil film pressure increases and the oil film load-carrying capacity increases, while the continuous increase of sliding velocity will reduce the adsorption capacity of lubricating oil between the walls, reduce the volume fraction of oil phase, and reduce the oil film load-carrying capacity. The results show that the sliding velocity has different effects on the loading capacity under different texture widths and depths. In addition, the influence trend of sliding velocity on the load-carrying capacity varies when the dimensionless texture depth and width are equal, but the film thickness difference is different ($\Delta h = 0.4$ μm and $\Delta h = 0.2$ μm), showing that the influence law of sliding velocity on the bearing performance is also affected by the film thickness difference.

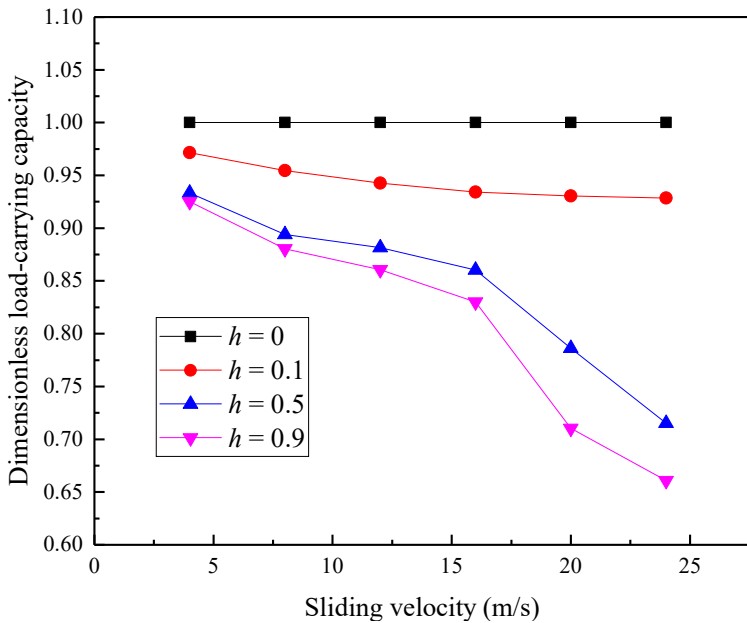

**Figure 12.** Variation of dimensionless loading capacity with sliding velocity at different texture depth ($\Delta h = 0.4$ μm).

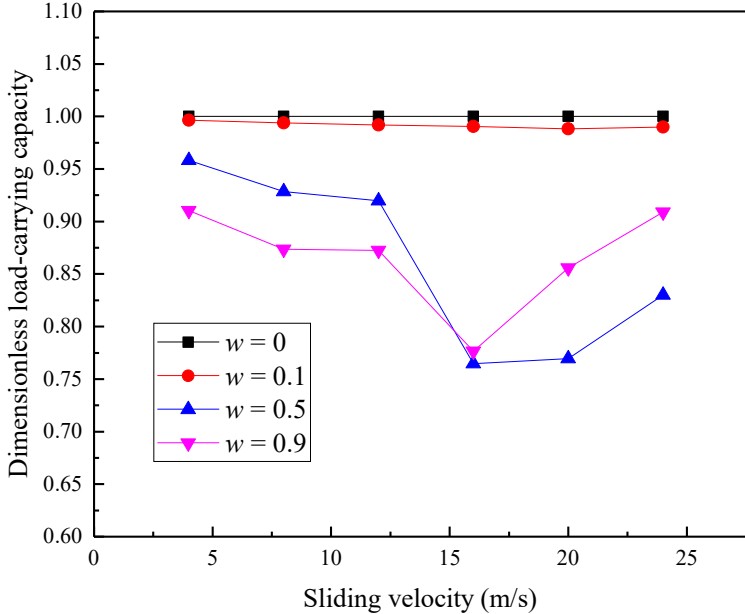

**Figure 13.** Variation of dimensionless loading capacity with sliding velocity at different texture width ($\Delta h = 0.2$ μm).

## 5. Conclusions

In this study, FLUENT is used to solve the 2D textured surface under convergent gap, and the convergence ratio parameter is studied to verify its irrationality in characterizing the load-carrying capacity of a textured surface. The film thickness difference and oil film thickness affecting the convergence ratio are independently studied, and the effects of these two parameters on the friction characteristics of non-parallel textured surface under the condition of convergence are considered. In addition, the effects of texture depth, texture width, and sliding velocity on the tribological properties of textured surface were analyzed. The conclusions are as follows:

(1) The same convergence ratio is constructed by changing the oil film thickness and film thickness difference. Although the oil film load-carrying capacity in the two cases has the same change trend, the change degree is different. The change of the load-carrying capacity with the convergence ratio varies when the film thickness difference or oil film thickness is different. Therefore, the convergence ratio is unreasonable in characterizing the friction performance of the textured surface.

(2) The effect of film thickness difference and oil film thickness on the tribological properties of textured surface under convergent gap was analyzed. When the film thickness difference is small, the introduction of texture will increase the load-carrying capacity of the surface. The cavitation effect gradually decreases with the increase in the film thickness difference, and the load-carrying capacity of texture surface will increase. The load-carrying capacity of textured surface decreases with the increase in the oil film thickness. Moreover, the textured surface shows better tribological properties when the oil film thickness is thick.

(3) The effects of texture depth, texture width, and sliding velocity on textured non-parallel surfaces were also studied. When the film thickness difference is large, the load-carrying capacity decreases with the increase in texture depth. When the film thickness difference is small, the load-carrying capacity first decreases, then increases, and finally decreases with the increase of texture depth. The load-carrying capacity first decreases, then increases, and finally decreases with the increase in texture depth. The texture width is optimal when the size of texture element is large, resulting in the highest oil film load-carrying capacity and texture effectiveness. When texture width and depth are small, the sliding velocity has little effect on the load-carrying capacity. When the texture width and depth are large, the load-carrying capacity will decrease with the increase in texture width. The influence of sliding velocity on the load-carrying capacity is also affected by the film thickness difference.

**Author Contributions:** G.B.: Conceptualization, Methodology, Investigation, Writing—review and editing. C.M.: Writing—original draft, Software, Validation, Supervision, Funding acquisition. X.W.: Data curation. J.S.: Validation, Data curation, Funding acquisition. X.N.: Formal analysis. All authors have read and agreed to the published version of the manuscript.

**Funding:** National Key R&D Program of China (2018YFB2000800) and the National Natural Science Foundation of China (52075268).

**Data Availability Statement:** Not applicable.

**Conflicts of Interest:** The authors declare that they have no known competing financial interests or personal relationships that could have appeared to influence the work reported in this paper.

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
