# Peer review of "Study on Tribological Characteristics of Textured Surface under Convergent Oil Film Gap"

_lubricants, doi:10.3390/lubricants10080183_

Round 1
Reviewer 1 Report
In this paper, the authors established 2D model of textured non-parallel surface and studied the load-carrying capacity of the textured surface by FLUENT. The influence of various parameters of textured surface on the load-carrying capacity were investigated, such as tilt angle, oil film thickness, texture depth, texture width, and sliding velocity. The following issues or comments need addressing /implementing by the authors:
1. There are many recent review articles on surface texture and tribology, none of which is included in the introduction. Therefore, the technical level has not been well reflected at all. It is suggested that the relevant contents in the introduction can be expanded to a certain extent
2. Please make a simple revision to the abstract, put forward a brief motivation and introduction, and briefly explain how the main results are achieved. Then, the authors should present their main findings in a concise way
3. The caption of Table 4 and Table 5 should be improved.
4. The captions of the figures should be improved. They are rather short and unspecific.
5. Please reduce the conclusion appropriately to make it more concise and clear.
The tribological characteristics of textured surface under convergent oil film gap are studied in this paper. The structure of the paper is easy to understand. The proposed method is still useful in application, and the results and discussion are relatively complete. So I suggest that the article can be published after minor revision.
Round 2
Reviewer 2 Report
See attached comments.
